# Effect of Olive Cake in Bísaro Pig Feed on Physicochemical Composition and Fatty Acid Profile of Three Different Muscles of Dry-Cured Shoulder

**DOI:** 10.3390/ani14111697

**Published:** 2024-06-05

**Authors:** Ana Leite, Lia Vasconcelos, Sandra Rodrigues, Etelvina Pereira, Rubén Domínguez-Valencia, José Manuel Lorenzo, Alfredo Teixeira

**Affiliations:** 1Centro de Investigação de Montanha (CIMO), Instituto Politécnico de Bragança, Campus de Santa Apolónia, 5300-253 Bragança, Portugal; anaisabel.leite@ipb.pt (A.L.); lia.vasconcelos@ipb.pt (L.V.); srodrigues@ipb.pt (S.R.); etelvina@ipb.pt (E.P.); 2Laboratório para a Sustentabilidade e Tecnologia em Regiões de Montanha, Instituto Politécnico de Bragança, Campus de Santa Apolónia, 5300-253 Bragança, Portugal; 3Centro Tecnológico de la Carne de Galicia, Avd. Galicia N° 4, Parque Tecnológico de Galicia, San Cibrao das Viñas, 32900 Ourense, Spain; rubendominguez@ceteca.net (R.D.-V.); jmlorenzo@ceteca.net (J.M.L.); 4Área de Tecnoloxía dos Alimentos, Facultade de Ciencias de Ourense, Universidade de Vigo, 32004 Ourense, Spain

**Keywords:** native breed, Bísaro, olive cake, muscles, curing process, nitrite-free, dry-cured shoulder, chemical composition, fatty acid profile

## Abstract

**Simple Summary:**

The use of olive cake in animal feed can be utilized for purposes that go beyond nutritional needs, in the sense of a circular economy. This study has shown that the use of olive cake does not negatively affect the physico-chemical characteristics and fatty acid profile of the final product (shoulder), which allows us to utilize this highly environmentally harmful by-product. This study also looked at the effect of curing times (fresh, 6 and 12 months curing) and, as expected, curing time was highly significant. The effect of three different shoulder muscles was also studied, and there were different behaviours due to the more external or internal location of the muscles.

**Abstract:**

The purpose of this study was to assess the following effects: (1) the inclusion of olive in the animal’s diet on the dry-cured shoulder; (2) the effect of curing on three different muscles (infraspinatus, supraspinatus, and subscapular); (3) the effect of different curing times (fresh shoulder, 6 months curing, and 12 months curing). For this purpose, forty shoulders were used, followed by a cold nitrite-free curing process with controlled humidity and temperature, according to the flowchart of a company that implements highly stringent standards in terms of food safety and quality. Samples were evaluated for their physicochemical composition and lipid profile. All the physicochemical composition parameters were significantly different (*p* < 0.001) in the three muscles studied. As might be expected, the curing times from the fresh product to the final product were also significantly different for all the parameters studied in this work. Regarding the inclusion of olive cake, it was found that treatment with a base diet + 10% exhausted olive cake (T4) showed higher levels for the parameters NaCl, collagen, and total fat. As for the fatty acid profile, in general, the olive did not influence the final product. On the other hand, we found that the type of muscle and the curing time of the cured shoulder had a significant influence on the fatty acid profile. We should also point out that there are significant differences in the interaction between curing time and muscle, particularly in saturated fatty acids (SFAs), monounsaturated fatty acids (MUFAs), and polyunsaturated fatty acids (PUFAs), as well as the lipid quality indices. Canonical discriminant analysis is viable for evaluating the evolution of the curing process, discriminating and classifying curing times, and evaluating the muscles of the Bísaro pork shoulder. Also, the introduction of olive cake into the animal diet does not affect the final product obtained.

## 1. Introduction

The Bísaro breed is one of the three indigenous pig breeds in Portugal, and most of the pig farms are concentrated in the northeast of Portugal. The excellent quality of meat from the Bísaro breed and its incredible suitability for processing typical high-quality products may be the reason for its conservation and growing importance today, especially as the basis for processed products with PDO- and PGI-quality labels, which are of great importance to the regional economy [1,2]. In addition to the exploration of this breed of pig in the region of Trás-os-Montes, there is another industry with a high impact in this region: the olive oil sector. Olive cake is one of the low-value by-products of the olive oil extraction process. It is a by-product that is highly toxic, causing concern not only for society in general but also for the olive oil sector itself due to the difficulty of storing it and its toxicity to the soil.

The collaboration between these two prevalent elements in Trás-os-Montes, also widely seen across Europe, has the potential to yield mutual advantages for both domains. Concurrently, there is a growing awareness regarding the ecological viability of these production systems. In response to this, it is imperative to devise and implement strategies that align animal production with the tenets of a circular economy. This approach must be balanced with the continued development and preservation of indigenous livestock breeds that are on the brink of extinction. Incorporating this by-product into the feed of pigs of native breeds can be seen as a strategy for valorizing a by-product, stimulating the use of available resources, and investing in environmental sustainability. On the other hand, the introduction of olive cake into the diet of animals can influence the quality of the meat and its processed products and can be a possible alternative to typical/commercial base diets. In addition, it can improve the lipid quality of meat, reduce fat thickness, and provide more energy, as is the case with ruminants [3,4,5,6,7,8]. Depending on each country and its region, processed pork products have designations of origin, following very specific recipes that make them unique from region to region. 

Dry-cured products are characterized by the fact that they are treated with salt, followed by a curing process that consists of reducing the moisture levels in the product and enzymatic processes that allow for the development of organoleptic characteristics typical of this type of product [9]. Curing processes lead to changes in the chemical composition, water activity, color, and flavor of dry-cured products [10,11]. According to Toldrá, [12], the quality of cured products (such as shoulders and hams) depends on water loss, salt diffusion, and the evolution of enzymatic activity during the different stages of the curing process. Bísaro pork shoulders are not widely available in the European Union or even in Portugal, where it is mainly consumed in the Trás-os-Montes region. This breed is used to produce various processed products, such as “alheira” [13], “butelo” and “chouriça” [14] “azedo” [15], ham [13], etc. However, the dry-cured shoulder is not included in this, despite being a characteristic product of the region with excellent qualities that can make it a high-value product. The high level of acceptance by local consumers is because this product has unique characteristics due to the excellent raw material from Bísaro breed animals and the curing methods (nitrite-free) of innovative industries that never forget the past and maintain traditional aspects. 

During the curing process of this type of product, significant physicochemical changes occur that affect the aqueous content, water activity, and color, among others. The dehydration of the shoulder begins in the first stage of manufacture (salting) and continues in the following stages of production. However, the intensity and speed of this process depend on the protection of each of the muscles that make up the shoulder, as well as other variables (air velocity in the chamber, relative humidity, etc.). For this reason, the study of some of the muscles that make up this product is of the highest importance for understanding how the curing time influences the physicochemical composition and fatty acid profile of each muscle. 

Therefore, this work aimed to study the effect of the curing process on three different shoulder muscles (infraspinatus, supraspinatus, and subscapularis), as well as to determine the evolution of the product from the beginning of the curing process to the end of the curing process with the addition of olive cake to dry-cured Bísaro shoulder, and to determine the effect of the curing process on the physicochemical composition and lipidic profile. 

## 2. Materials and Methods

### 2.1. Animals and Diets

This study is based on animals of the autochthonous Bísaro breed, raised in an extensive system. This extensive system allows the animals to have larger areas in which to move around and explore, thus promoting animal welfare. In addition, they have access to a wide range of natural foods that the intensive system does not have. The forty animals used for this study were divided into five groups (eight animals in each group) on a farm owned by Bísaro-Salsicharia Tradicional, Lda^®^ in the north-east of Portugal (Gimonde-Bragança, Portugal). The first treatment (T1) consisted of animals fed a traditional diet for these autochthonous breeds in an extensive system and was considered the control treatment. Traditional diets typical of these breeds contain a series of vegetables and cereals, supplemented with a specific commercial feed for each stage of growth. In addition to the base diet typical of these animals, the remaining treatments included 10% of olive cake. All the olive cake used by these pigs comes from various olive mills in the Trás-os-Montes region. Treatments T2, T3, T4, and T5 differed in the way the olive cake was extracted. Treatment T2 consisted of the base diet + 10% crude olive cake; Treatment T3 consisted of the base diet + 10% olive cake, two phases; Treatment T4 consisted of the base diet + 10% exhausted olive cake, and treatment T5 consisted of the base diet + 10% exhausted olive cake + 1% olive oil. These types of olive cake extractions were selected because they are the most representative of this region. Only then would it make sense to make real use of this by-product. 

Feed was applied to all groups simultaneously and under the same conditions (feed level “ad libitum” with an average consumption of 3 kg per day). The experimental feed trial was carried out during the finishing phase, for a total of 3 months, at Trás-os-Montes e Alto Douro University, Vila Real, Portugal. Analysis of diets was performed at the Meat Technology Center of Galicia, Ourense, Spain. Animals were harvested when they reached 12 months of age, at around 135 kg live weight and 110 kg carcass weight, at the Municipal Slaughterhouse of Bragança. The slaughter and carcass preparation process were previously described by Álvarez-Rodríguez and Teixeira [16]. All animals were cared for and slaughtered in compliance with the welfare regulations and respecting EU Council Regulation (EC) No. 1099/2009 [17]. After 24 h of slaughter, the carcasses were taken to the cutting room to be obtained into different joints. After cutting, all the joints obtained were sent to the company Bísaro-Salsicharia Tradicional, Lda^®^. The joints obtained from the cutting process were transported and stored at a temperature ≤ 5 °C (monitored temperatures). Approximately 48 h elapsed between slaughter and the start of the shoulder production process.

Ingredient composition, chemical composition, and the fatty acid profile of each diet treatment are shown in Table 1.

### 2.2. Dry-Cured Bísaro Shoulder

Shoulders were processed and cured for 12 months in the “Bísaro-Salsicharia Tradicional, Lda” company. Curing and drying were carried out through several stages, including salting (without added nitrites), post-salting (stabilization), drying, and ripening. For the salting stage, the shoulders were massaged with salt and kept in a cold room for 1 day per kg of fresh weight. The chamber temperature for the salting period was 0–3 °C and relative humidity was 85–90%. At the end of the salting stage, superficial salt was removed from the shoulders using pressurized warm water. During the post-salting step, the shoulders were kept for 90 days at 0–5 °C and at a relative humidity of 80–85%. After achieving stabilization, the shoulders were moved to a chamber (drying step), where the temperature was gradually increased from 8 to 16 °C and the relative humidity was dropped to 75–80% (for 4 months). The shoulders were moved to another chamber (ripening step), where the temperature was gradually increased from 16 to 30 °C at a relative humidity of 65–68% (for 3 months). A flowchart of dry-cured Bísaro shoulder processing is presented in Figure 1.

Laboratory analyses were carried out on the fresh shoulder (after cutting the carcass—RMC) and during two different curing periods. The first curing period concerns the stage when the product has completed 6 months of the manufacturing flowchart (halfway through the curing process—6MC). The second curing period concerns the final curing phase when the product is already destined for commercialization (12MC). In addition, within these three curing times (RMC; 6MC; 12MC), three different muscles were removed from the shoulder (infraspinatus—IF, supraspinatus—SP, and subscapular—SB). 

### 2.3. Chemical Composition and Physicochemical Analysis 

The chemical composition of the raw and dry-cured shoulder (6 and 12 months of curing) was analyzed using established procedures. The determination of moisture was performed according to the Portuguese Institute of Quality (NP 1614/2002) [18]. In duplicate, 3 g of the sample was added to 5 mL of ethanol (96% *v*/*v*). After that, samples were dried in a drying oven (Raypa DO-150, Barcelona, Spain) for 24 h at 103 ± 2 °C. Ashes were assessed according to the Portuguese Institute of Quality (NP 1615/2002) [19]. To 3–5 g of the sample, we added 1 mL of magnesium acetate (15% *w*/*v*) in crucibles. After that, the samples were subjected to 550 °C ± 25 °C for 5–6 h in a muffle furnace (Vulcan BOX Furnace Model 3–550, Yucaipa, CA, USA). Protein determination was carried out following the Portuguese Institute of Quality (NP 1612/2002) [20] using the Kjeldahl Sampler System (K370, Flawil, Switzerland) and the Digest System (K-437, Flawil, Switzerland). Two grams of the sample were put in mineralization tubes with two catalyst tablets and twenty-five milliliters of sulfuric acid (97%). After mineralization completion, the distillation procedure was carried out. Finally, the distillate was titrated with hydrochloric acid solution and the required volume was recorded. All parameters were expressed in percentage (g/100 g of product). Water activity (aw) was determined using a water activity probe (HygroPalmAw1 rotronic 8303, Bassersdorf, Switzerland) according to AOAC International [21]. The hydroxyproline determination of the collagen content and concentration was determined according to the methodology recommended by the Portuguese Institute of Quality (NP 1987/2002) [22]. The total chloride content was analyzed according to the recommended methodology in the Portuguese Institute of Quality (NP 1845/1982) [23].

### 2.4. Fatty Acid Analysis

Fatty acids in the shoulders of Bísaro pork were analyzed at the Carcass and Meat Quality Laboratory of ESA-IPB following the method proposed by the Folch procedure [24]. The total lipids were extracted from 25 g of the meat sample. The fatty acid profile was determined using 50 mg of fat. The fatty acids were transesterified according to the method described by Domínguez et al. [25]; after adding 4 mL of a sodium methoxide solution and vortexing it for five minutes at a time for 15 min at room temperature, 5 mL of the H_2_SO_4_ solution (in methanol at 50%) was added. Then, 2 mL of distilled water was added, and the samples were vortexed again. The organic phase (with the methyl esters of fatty acids) was extracted with 2.35 mL of hexane. The fatty acid methyl esters separation and quantification were performed using a gas chromatograph (GC-Shimadzu 2010Plus; Shimadzu Corporation, Kyoto, Japan) provided with a flame ionization detector and an automatic sample injector AOC-20i and using a Supelco SP TM-2560 fused silica capillary column (100 m length, 0.25 mm i.d, 0.2 µm film thickness). The fatty acid contents were calculated using chromatogram peak areas and were expressed as g per 100 g of total fatty acid methyl esters. In addition, the percentage of saturated fatty acids (ΣSFA), monounsaturated fatty acids (ΣMUFA), and polyunsaturated fatty acids (ΣPUFA), the ratio PUFA n-6/n-3, and Σtrans were calculated according to Vieira et al. [26]. To measure the lipid quality, the atherogenic index (IA) and thrombogenic index (IT) were calculated according to Ulbricht and Southgate [27]. 

### 2.5. Statistical Analysis

Data were tested for normal distribution and homogeneity of variance using the Shapiro–Wilk test. Next, the effect of treatment, curing time, and muscle, and the interaction between curing time and muscle on the chemical composition and fatty acid profile were examined using analysis of variance (ANOVA) with the general linear model (GLM) procedure, in which these parameters were defined as dependent variables and curing time, treatments, and muscle as fixed effects. The results were presented in terms of mean values and standard error of the mean (SEM). When there was a significant effect (*p* < 0.05), the means were compared using Student’s *t*-test. To find out which group of chemical composition variables are most useful for classifying and distinguishing the nine shoulder groups, a discriminant analysis was carried out using the linear, common covariance, and stepwise variable selection methods (PROC DISCRIM, SAS). The effectiveness of the discriminating power of the selected models was assessed using Wilks’ lambda value test. The results were analyzed in terms of the absolute assignment of individuals to the pre-assigned group and the variance explained by each canonical similarity, as well as by analyzing the score coefficients. To extract a few key combinations (called principal components) from the group of measured variables that capture most of the variability in those variables, we conducted a principal component analysis (PCA). Each principal component was determined by combining the eigenvectors of the correlation matrix linearly. The eigenvalues indicate how much variance each component holds. Also, a multiple factor analysis (MFA) related to principal components analysis (PCA) was performed to produce a table of eigenvalues, summary plots, and a consensus map. All analyses were performed using the statistical package JMP^®^ Pro 17.0.0 by 2023 SAS Institute Inc.© (Cary, NC, USA).

## 3. Results and Discussion

### 3.1. Influence of Olive Cake Treatments on Physicochemical Composition

The addition of olive cake to the diet of Bísaro pigs did not influence most of the physicochemical parameters evaluated in this study. It should be noted that, in the parameters where there was significance (NaCl, total fat, and collagen), treatment T4 stood out from treatment T5, obtaining higher values for the three parameters mentioned above. As reported in Table 2, olive cake had a significant effect on some parameters of the physicochemical composition, such as NaCl, collagen, and total fat. For these parameters, treatment T4 obtained the highest value, being significantly different from treatment T5 (for NaCl and total fat) and treatment T2 (for collagen). The other parameters were not significantly influenced by the addition of olive cake to the animal diet. In similar processed products (dry-cured loin and dry-cured “cachaço”), with a shorter curing time, lower NaCl values were obtained, and there were no significant differences between the treatments with 10% of olive cake [28]. In another study on the inclusion of olive cake in the diet of Bísaro pigs [28], similar collagen values were obtained for the dry-cured loin (1.59–2.84%) and the dry-cured “cachaço” (1.54–2.09%). Higher NaCl values were obtained for dry-cured Celtic pork ham, which was not fed olive cake [29]. However, lower values were obtained for dry-cured shoulders from different Iberian pig genetic lines, with values between 3.5 and 4% [30]. For total fat, values between 11 and 14% were obtained for dry-cured shoulder. As mentioned above, the olive cake had a significant influence on treatment T4 and T5. For the same type of product, other authors have obtained lower total fat values of around 8% [30]. According to Delgado et al. [31], higher total fat values were obtained for dry-cured shoulders from Mexican pigs. For 180 days of dry-curing, average values of 20% were obtained. Similar values for total fat (11%) were obtained for Iberian dry-cured shoulder with a change in the animal’s diet [32]. 

The other parameters studied (protein, ash, moisture, and aw) were not significantly affected by the incorporation of olive cake into the animal’s diet. For protein, values between 26.54 and 27.56% were obtained, which is in line with the values obtained in the dry-cured shoulder from different genetic lines of the Iberian pig [30]. On the other hand, higher values were obtained in dry-cured shoulders from Mexican pigs, at around 30% protein, whose curing time is shorter [31]. Higher protein values were also obtained for the Teruel dry-cured ham, at around 31% [33]. Regarding the percentage of ash, the values obtained in this study range from 4.35 to 4.88% for all the treatments applied to the animal’s diets. Other authors have obtained average ash values of 8.13% for dry-cured shoulder [30] and Teruel dry-cured ham (6.58–7.02%) [33]. Moisture in the Bísaro dry-cured shoulder reached an average final value of between 53 and 56%, with a similar behavior for all the treatments with olive cake applied to the animal’s diet. According to other authors [30], Iberian dry-cured shoulder obtained average values of between 51 and 56%, like those obtained by us. The average aw values obtained for this product varied between 0.896 and 0.899 for all the treatments studied, with the olive cake not influencing the final water activity value of the dry-cured shoulder. Aw values control the product’s capacity for contamination and influence the stability of the meat. Aw values below 0.90 inhibit pathogens such as *clostridium.* Similar values were obtained in dry-cured Mexican shoulder [31], dry-cured Iberian shoulder [30], and dry-cured Celta ham [29].

### 3.2. Influence of Curing Time and Type of Muscle on Physicochemical Composition

Table 3 shows the three muscles studied and the curing times applied to Bísaro dry-cured shoulder. In the case of water activity (aw), as expected, we can see that, over the curing time, all the muscles significantly decrease (*p* < 0.001) their value from the beginning of the curing time (fresh product) to the finished product (12 months of curing). The aw obtained for fresh muscles varies between 0.954 and 0.963, which is characteristic of fresh meat. The initial aw values are in line with the values obtained for fresh meat by other authors [34,35,36,37]. For the intermediate curing time, the values obtained for the three muscles studied vary between 0.915 and 0.896, with no significant differences for this curing time (*p* > 0.05). Previous studies by other authors [30,38] on Iberian dry-cured shoulder (from various crosses) with a curing period of 240 days reported values similar to those observed in our study after 6 months of curing. Similarly, in the case of Celtic dry-cured ham [29], aw values comparable to those observed in our study during the intermediate curing phase were obtained during the initial curing phase (1ª “bodega”) of the ham. This shows that, from the beginning of the processing, there is a high decrease in water activity, which is related to the incorporation of salt. As noted in Table 3, the subscapularis (SB), infraspinatus (IF), and supraspinatus (SP) muscles had the same effect when it came to lowering aw. The SB muscle, being the outermost muscle, obtained the lowest value in the two curing times studied. The value obtained for this muscle at 12 months was significantly lower (*p* < 0.001) than for SP and IF. Since SB is an external muscle, it is not covered by the fat covering, which means that it quickly develops a low water content in the early stages of curing, which is the cause of the value obtained in this study for this muscle. According to the other authors [37], the Bísaro dry-cured shoulder aged for 12 months exhibited higher water activity (aw) values compared to any of the muscles analyzed in our study. Similarly, the curing process resulted in a reduction in the moisture content of the muscles analyzed, as observed with water activity. As with water activity, the curing process also caused a decrease in the moisture content of the muscles studied. The values obtained in this work for water activity are very positive since values below 0.90 inhibit pathogens such as *clostridium*. The moisture content for each of the fresh muscles varied between 71.10 and 72.19%, with no significant difference between them (*p* > 0.05). However, following salting, post-salting, and the start of drying, each muscle displayed distinct moisture values. The SP muscle has the highest moisture content (59.28%), followed by the IF muscle (52.75%). Because it is a more external muscle, the SB muscle is the one that dehydrates the fastest, obtaining significantly lower (*p* < 0.001) moisture values, at around 42.27%. At the end of the curing process, we have significantly different (*p* < 0.001) values between the muscles studied. As with water activity, the muscle with the highest water content was SP, with a value of 50.32%. Next was the IF muscle, with a moisture content of 43.99%, and, lastly, the muscle with the lowest moisture content in the final process of curing was the SB muscle, with an average value of 33.24%. This effect has also been verified by other authors in a similar product [29]: dry-cured ham. In dry-cured ham, final moisture values of 52.35% and 35.82% were obtained for the internal muscles (*biceps femoris*) for the external muscle (*semimembranosus*), respectively [29]. In a study on dry-cured Iberian pork shoulder [30], final product moisture values ranging between 51 and 56% were reported, which is close to the moisture content observed in the internal muscles after 6 months of curing time. Other researchers [39] observed higher total moisture content values, with an average of 46.34%, for the dry-cured shoulder. For Mexican dry-cured shoulder [31], average values of 31% were obtained, close to the values that we obtained for the SB muscle of Bísaro pork shoulder. Regarding the amount of salt, we observed a different trend for the three muscles studied. The SB muscle obtained a lower NaCl value for the intermediate and final curing times when compared to the SP and IF muscles. The SB muscle obtained final values of 3.81%, with no significant differences when compared to the intermediate curing time, which obtained an average value of 3.75%. The IF and SP muscles showed the same effect between 6 and 12 months of curing. The curing time of 6 months for the IF and SO muscles obtained an average value of 4.32% and 4.96%, respectively. For the 12-month curing time, the IF and SP muscles obtained average values of 6.09% and 6.13%, respectively. The observed values for the SB muscle are likely influenced by its direct exposure to salt. There was probably a rapid uptake of salt during the salting phase, given this muscle’s direct contact with the curing salt. Following this initial phase of maximum values, the salt content gradually decreased throughout the curing process. In IF and SP muscle, the salt is distributed more evenly, causing the salt concentration to increase throughout the curing process. The position of the muscles significantly influences the distribution of salt during the curing process. Other authors have reported similar conditions in different muscles of Celtic dry-cured ham [29]. The external muscle, with direct contact with the salt during the salting stage, obtained maximum values during the salting stage, after which the salt content decreased until the end of the curing process [29]. According to Reina et al., [39], final NaCl values for dry-cured shoulder were 8.19%, which is much higher than the value we obtained. According to Leite et al., [37] average values of 4.12% were obtained for the salt content of Bísaro dry-cured shoulder (where all the muscles of the dry-cured shoulder are involved). The ash values are in line with the trend observed in the NaCl content, increasing significantly throughout the curing process. As expected, the higher the NaCl content, the greater the amount of ash obtained. The SB muscle obtained a final value of 5.38%, the IF muscle obtained a final value of 7.08%, and the SP muscle obtained a final value of 7.15%. As was the case with the salt content, ash increased the most in the early stages of the curing process. 

A significant increase (*p* < 0.001) in the protein content was observed throughout the curing process due to dehydration caused by the addition of NaCl. Initially, all the muscles had a protein content of around 19%, with no significant differences between the various muscles studied. As the curing process advanced, we observed significant variations in this parameter between the muscles. The subscapularis muscle showed the highest value for this parameter, followed by the supraspinatus and finally the infraspinatus. The protein values obtained for Bísaro dry-cured shoulder for 12 months of curing are in line with the average values also obtained for Teruel dry-cured ham [33], Mexican dry-cured shoulder [31], and Bísaro dry-cured shoulder [37]. As for the total fat content, we can see that the fresh muscles had average values of between 6.34 and 7.99%, with no significant differences between them. However, as with the other parameters already discussed, the dehydration of the muscle during the curing process will significantly increase the fat content. The average value obtained at the end of the curing process varied between 23.02% for the SB muscle, 13.97% for the IF muscle, and 12.06% for the SP muscle. As dehydration occurred, leading to higher levels of dry matter, the Bísaro dry-cured shoulder muscles exhibited increased protein, total fat, and ash content. Except for moisture, the other parameters of the chemical composition and the NaCl content (expressed in g/100 g of fresh product) increase with the stages of the curing process, which is in line with other studies [35,36,40]. Collagen showed significant differences (*p* < 0.001) during the drying process. The fresh muscles obtained values that were not significantly different from each other, ranging from 0.94 to 1.64%. For the SB and SP muscles, there was a decrease in collagen in the first stages of curing, while the same trend was not seen in the IF muscle, with the same values for the fresh muscle and the muscle after 6 months of curing. At the end of curing (end of first drying and second drying), all the muscles had significantly increased collagen content. The SP muscle obtained a significantly lower value (2.48%) when compared to the IF and SB muscles. There were no significant differences between the IF and SB muscles, with values of 4.36% and 3.62%, respectively. 

### 3.3. Influence of Olive Cake Treatments on Fatty Acid Profile

Table 4 shows the effect of adding olive cake on the fatty acid profile of the Bísaro pork shoulder. The inclusion of olive cake in the finishing diet of these animals had no significant influence on the profile’s fatty acids, nor their fractions and indexes. It should be noted that significant differences were observed in the most abundant fatty acid (oleic acid). Treatment T2 had a lower value for this fatty acid (47.76%), while treatment T3 had the highest value (48.78%). The same applied to other products, such as dry-cured loin and dry-cured “cachaço” [28], obtained from animals fed a finishing diet with olive cake, where the fatty acid profile of Bísaro pork shoulder was not influenced either. For all the treatments with the inclusion of 10% olive cake with different extraction methods and for the control treatment (without the addition of olive cake), the most abundant fatty acids were palmitic acid (C16:0) as saturated fatty acids (SFAs), oleic acid (C18:1n-9) as monounsaturated fatty acids (MUFAs), and linoleic acid (C18:2n-6) as polyunsaturated fatty acids (PUFAs). Therefore, the sum of these majority fatty acids (C18:1n-9; C16:0; C18:0; and C18:2n-6) represent more than 90% of the total fatty acids of Bísaro pork shoulders with the introduction of olive cake into the animal’s diet. The most abundant fatty acids in the dry-cured shoulder were MUFAs, followed by SFAs and PUFAs, which is in line with the fatty acid profile typical of meat pork. 

### 3.4. Influence of Curing Time and Type of Muscle on Fatty Acid Profile

Table 5 shows the fatty acid profile of the Bísaro dry-cured shoulder with three curing times (raw meat, 6 months curing, and 12 months curing) in three different muscles (subscapularis (SB), infraspinatus (IF), and supraspinatus (SP)). The most abundant fatty acids in the dry-cured shoulder were MUFAs, followed by SFAs and PUFAs, which is in line with the fatty acid profile typical of pork meat. This typicality is seen in other products from other breeds, such as dry-cured loin and “cachaço [28], Celta lacón [41], Iberian dry-cured shoulder [30,39,42], and hams [39,43]. 

Most of the fatty acids for the three muscles studied were C18:1n-9; C16:0; C18:0; and C18:2n-6. These individual fatty acids account for more than 90% of the total fatty acids. For oleic acid, the SB, IF, and SP muscles obtained values between 47 and 78%, with no significant difference between them at the fresh muscle stage. At the first curing time (6MC), a significant increase in oleic acid was observed for the SB and IF muscles. However, there was no significant change in the SP muscle. At the final curing (12MC), oleic acid decreased significantly in the SB muscle, remaining stable in the other muscles. The product exhibited oleic acid concentrations ranging from 46.57% to 48.54%, with the SB muscle displaying the lowest levels of this fatty acid. Palmitic acid was identified as the second most prevalent fatty acid in the product. The SB muscle was the one that obtained the most accentuated increases in this fatty acid throughout the curing process. Values of 23.73% and 26.01% were obtained in the fresh meat and final product, respectively. Regarding the IF and SP muscles, the values were very similar for the final product, between 24,73% and 24,69%, respectively. The SB and SP muscles obtained a significantly higher (*p* ≤ 0.001) value of this acid in the final curing process. Stearic fatty acid is the third most abundant fatty acid in all the muscles studied in this research. The curing process led to an increase in stearic fatty acid in SB and SP muscles. Regarding the IF muscle, there was a significant decrease in this acid between the fresh product and the 6M curing time. The final product obtained a value like that obtained in the fresh product (around 11.8%). The internal muscles SP and IF obtained identical final values for this acid (11.77% and 11.78%, respectively). On the other hand, the external SB muscle obtained significantly higher values (12.77%) of this acid when compared to the other muscles. In terms of linoleic acid, the interaction between muscle and curing time was significant, as was the case with the other major fatty acids. In this case, all the muscles had the same tendency, with a decrease in this fatty acid throughout the curing process. The final value of this acid for the SB, IF, and SP muscles was 6.89%, 7.84%, and 8.47%, respectively. That said, these four major fatty acids will significantly influence the SFA, MUFA, and PUFA fractions.

During curing, saturated fatty acids (SFAs) increased in the SB and SP muscles, reaching a final value like that of the fresh shoulder. In another study [37] using Bísaro pork shoulder without muscle separation, researchers observed a non-significant rise in SFAs during curing, with final average values of 37.20%, consistent with the values found in the IF and SP muscles. Despite these differences, it should be emphasized that the difference is less than 2% and is related to the fact that this muscle is more external and therefore suffers greater and faster dehydration. The other authors [32] obtained average SFA values of 39% for dry-cured Iberian shoulder with a change in the animal’s diet. On the other hand, other dry-cured Bísaro products [28], such as dry-cured loin and “cachaço”, had higher SFA values (42% and 43%, respectively). In the total MUFA content, we observed a significant difference in the interaction between muscle and curing time. There was a significant increase in this fraction in all the muscles studied from the start of curing (RM) to the middle of the curing process (6MC). On the other hand, the value obtained in the final product (for all muscles—12MC) is not affected by curing time. This can be an important nutritional aspect since MUFA reduces cardiovascular risk factors [44]. Moreover, MUFA reduces plasma LDL cholesterol levels without impairing the anti-atherogenic properties of HDL cholesterol lipoproteins [45]. The muscle–curing time interaction significantly impacted the PUFA fraction. Across all muscles studied, the PUFA content consistently decreased from the beginning to the end of the curing process. In the final product, the SB muscle had the lowest PUFA content at an average of 7.58%, followed by the IF muscle at 8.68%, while the SP muscle showed the highest PUFA content at 9.28%. During curing, PUFAs are more susceptible to oxidative degradation compared to SFAs and MUFAs, leading to their conversion into secondary molecules [46]. All the muscles obtained a higher n-6/n-3 ratio for the final product, with the SB muscle obtaining final values of 28.06%, the SP muscle 26.89%, and the IF muscle 26.13%. The value of this ratio for dry-cured shoulder exceeds the internationally recommended values (<4) for a healthy and balanced diet [47,48], the ideal value being 1 [49,50,51]. Lower values (16.08%) of this ratio were found for dry-cured shoulders of the same native breed (without muscle division) [37] and dry-cured shoulders and dry-cured ham [39] from white-breed pigs. Similar values were found for other processed products from Bísaro pig, such as dry-cured loin and dry-cured “cachaço” [28].

The index of atherogenic (IA) and the index of thrombogenic (IT) characterize the atherogenicity and thrombogenic potentials of fatty acids, respectively [27]. The interaction between muscle and curing time was significant for the IA and IT indices. Through these indices, we can see that the curing process (in all the muscles studied) increased these indices. It should be noted that the muscle that undergoes a faster healing process due to its more external location (SB) obtained more negative values for these indices. However, to date, no organization has provided recommended values for this index [52]. The h/H ratio refers to the functional effects of fatty acids on cholesterol metabolism, and Cava et al. [53] report that the higher this ratio, the more nutritionally suitable the fat in the food. As with the IA and IT indices, the h/H index obtained a final value for the IF muscle very similar to that obtained for the fresh shoulder. This muscle, being the internal one, does not see these indices negatively affected in the curing process. On the other hand, the most external muscle (SB) significantly decreases this index.

### 3.5. Discriminant Analysis

Discriminant analysis is a powerful descriptive and classificatory technique for describing specific characteristics of distinct groups and classifying cases into pre-existing groups based on the similarities between that case and the other cases belonging to the groups [54]. The *F* values of all variables considered in the discriminant analysis to determine if the nine shoulders x curing time groups could be distinguished based on the chemical composition, a fraction of saturated fatty acids, and n-6/n-3 ratio are shown in Table 6.

A scatter plot of the first two canonical variables of the nine groups considered (Figure 2) showed that groups were discriminated with a total of 92.70% of variance explained, with 83.86% and 8.84% for canonical 1 and canonical 2 functions, respectively. The model accepts a third significant canonical variable, explaining 4.75% of the total variance. It accepts fourth, fifth, and sixth significant (*p* < 0.001) canonical variables, explaining 1.53%, 0.72%, and 0.23%, respectively. The application of linear discriminant analysis showed that the use of eight variables made it possible to discriminate between the groups of shoulder muscles with different curing times. Figure 2 shows that the IF, SP, and SB muscles of the fresh shoulder are grouped, with no differences in their chemical composition. As the curing process progresses, the external SB muscle is isolated from the others, as can be seen in Figure 2. As mentioned before, this muscle suffers a high degree of dehydration in the very first stage of the manufacturing flowchart. This muscle does not have a layer of fat or skin, which means that, during the salting phase, the salt penetrates quickly without any barrier. As a result, the subsequent stages are much quicker, which makes the muscle significantly different in its composition. For this reason, we see that the SB muscle at the intermediate and final curing times is mostly positioned in the positive part of the second canonical variable. At curing times T6 and T12, the SP and IF muscles are very evenly distributed. The discriminant function becomes an important traceability tool for the food industry. This information is of great interest to the meat processing industry because it guarantees the type of product that they want to produce. In addition to what has already been mentioned, this information is extremely important for standardized industries with demanding quality standards of food. Obtaining certification for quality standards requires the processing sector to have traceability systems with very high standards so that the information provided to the consumer is as clear and precise as possible. As this is a highly heterogeneous matrix, this type of tool is essential for validating the product’s nutritional declarations so that the consumer is not misled. Therefore, this study demonstrates the viability of canonical discriminant analysis to evaluate the evolution of the curing process by discriminating and classifying the curing times and muscles of the Bísaro pork shoulder.

### 3.6. Principal Component Analysis

Principal component analysis is a statistical method that allows a set of case data to be reduced by variables to their essential characteristics, known as principal components. The principal components are a set of linear combinations of the original variables that maximize the variance of all the variables. The main graphical result often takes the form of a biplot, using the principal components to map the cases and adding the original variables to support the interpretation of the distance between the positions of the cases [55].

Figure 3 shows the results of the principal component analysis. The graphical result obtained takes the form of a biplot. Two significant eigenvalues were obtained (*p* < 0.001). The first principal component explained 65.2% of the total variance, and the second principal component explained 12.9% of the total variance (78.1% of the total variance). The positive region of the first factor (or principal component PC1) is mainly represented by samples that have already undergone a curing process (6MC and 12MC). The negative region of the first factor mainly shows the fresh pork shoulder samples. On the other hand, the positive region of the second factor (or PC2) is mainly represented by the SP and IF muscles with curing times of 6 and 12 months and part of all the muscles at the fresh shoulder stage. This is because the curing process in these two muscles (SP and IF) contains a higher salt content and consequently a higher ash content. In the case of the fresh product, this is because, as is to be expected, these samples contain a higher moisture content and aw. The negative region of component 2 mainly represents the SB muscle for the 6 and 12 months of curing and for most of the fresh samples of all the muscles. This is because the curing process in the SB muscle has a higher protein, collagen, and total fat content at both curing times.

The principal component analysis reinforces the data obtained in this study. The SB muscle, because it is an external muscle, has a different evolution throughout the curing process when compared to the IF and SP muscles.

## 4. Conclusions

Including 10% olive cake in animal diets, similar to other studied products, adds value by utilizing an olive industry co-product with significant environmental impact. This study focused on major muscles in Bísaro pork shoulder, revealing distinct curing processes and resulting in varied chemical compositions among the muscles. Differences in fatty acid profiles underscored the unique exposure of shoulder muscles to the curing process. As expected, the product undergoes significant differences throughout the curing process regarding its chemical composition and fatty acid profile.

In summary, it can be reaffirmed that the incorporation of olive cake into the animal diet does not affect the final product. Moreover, this by-product, despite its environmentally high toxicity, can be effectively utilized, thereby fostering a sustainable circular economy. Furthermore, the study of the musculus (supraspinatus, infraspinatus, and subscapularis) and the evolution of the curing time will provide characterization data for the Bísaro pork shoulder, contributing to establishing a quality label with significant added value. Additionally, this study provides the food industry with valuable tools for crafting a variety of products boasting excellent chemical properties.

## Figures and Tables

**Figure 1 animals-14-01697-f001:**
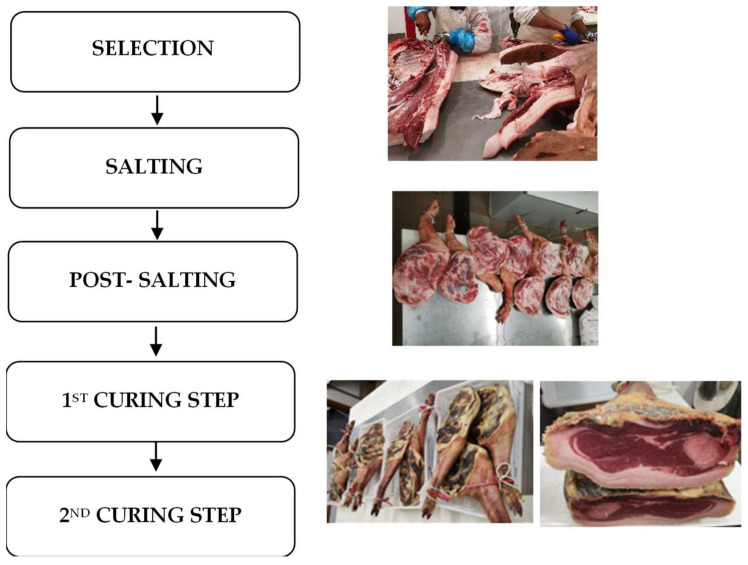
Flowchart of the fabrication of dry-cured shoulder of Bísaro pig.

**Figure 2 animals-14-01697-f002:**
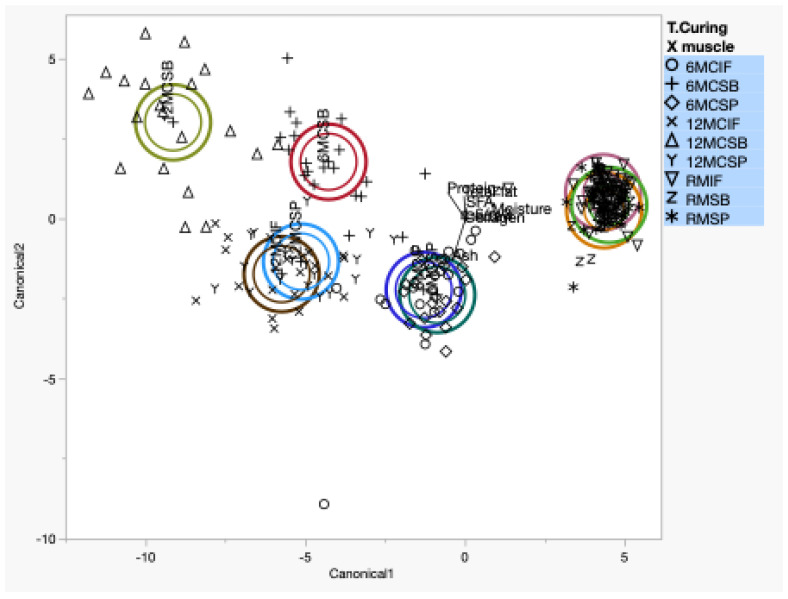
Scatter plot of the first two canonical variables of the nine groups considered (RMIF, RMSB, RBSP, 6MCIF, 6MCSB, 6MCSP, 12MCIF, 12MCSB, 12MCSP).

**Figure 3 animals-14-01697-f003:**
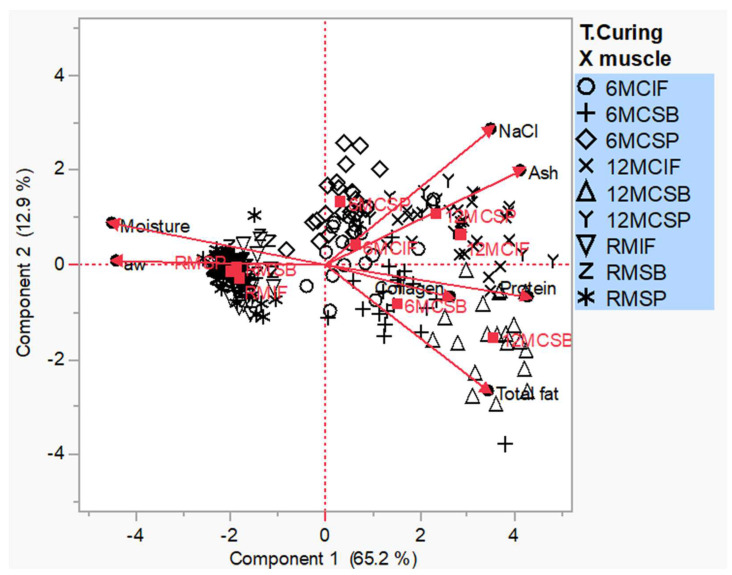
Biplot principal component analysis.

**Table 1 animals-14-01697-t001:** Ingredient composition of the experimental diets (g/kg, as fed basis) and fatty acids composition (g/100 g).

			Diets		
	T1	T2	T3	T4	T5
Olive cake	0	10	10	10	10
Olive oil	0	0	0	0	1
Barley grain	45.80	41.20	41.20	41.20	41.20
Wheat grain	22.60	20.40	20.40	20.40	20.40
Soybean meal 47	12.90	11.60	11.60	11.60	11.60
Rice bran	5.00	4.50	4.50	4.50	4.50
Corn grain	2.50	2.20	2.20	2.20	2.20
DDG’s corn	5.00	4.50	4.50	4.50	4.50
Beet molasses	4.00	3.60	3.60	3.60	3.60
Minerals and vitamins	1.70	1.70	1.70	1.70	1.70
Supplement min + vit + fitase	0.50	0.30	0.30	0.30	0.30
Chemical composition of the diet					
DM	98.05	98.49	98.19	98.15	98.46
OM	93.90	94.20	93.75	94.16	93.98
NDF	18.01	23.39	22.97	24.04	22.88
ADF	6.40	10.62	10.48	10.50	10.06
ADL	0.89	3.06	2.81	3.09	2.86
Cellulose	5.51	7.56	7.68	7.41	7.20
PB	16.00	13.38	13.45	14.39	13.98
GB	5.41	5.53	4.96	4.30	5.20
Fatty acids (g/100 g)					
ΣSFA	21.54	20.03	20.09	21.31	20.57
ΣMUFA	28.00	42.36	38.31	29.45	36.10
ΣPUFA	50.46	37.61	41.60	49.24	43.34
PUFA/SFA	2.34	2.07	1.88	2.31	2.10
n-6/n-3	17.58	16.66	16.69	17.10	16.90

DM—dry-matter; OM—organic matter; NDF—neutral detergent fiber; ADF—acid detergent fiber; ADL—acid detergent lignin; PB—crude protein; GB—crude fat. T1—base diet and commercial feed; T2—base diet + 10% crude olive cake; T3—base diet + 10% olive cake, two phases; T4—base diet + 10% exhausted olive cake; T5—base diet + 10% exhausted olive cake + 1% olive oil.

**Table 2 animals-14-01697-t002:** Effect of inclusion of olive cake to the animal diet on the physicochemical composition of Bísaro dry-cured shoulder.

Chemical Composition (g/100g)							
	aw	Moisture	NaCl	Ash	Protein	Total Fat	Collagen
T1	0.899 a	55.76 a	4.91 ab	4.88 a	27.56 a	11.74 ab	1.99 ab
T2	0.897 a	54.81 a	4.99 ab	4.57 a	26.93 a	11.08 b	1.65 b
T3	0.899 a	55.38 a	4.53 ab	4.35 a	26.69 a	11.62 ab	1.96 ab
T4	0.899 a	53.98 a	5.32 a	4.78 a	27.48 a	14.00 a	2.48 a
T5	0.896 a	56.19 a	4.46 b	4.36 a	26.54 a	11.00 b	1.82 b
SE	0.003	0.73	0.20	0.14	0.40	0.73	0.17
*p* value	ns	ns	*	ns	ns	*	**

T1—base diet and commercial feed; T2—base diet + 10% crude olive cake; T3—base diet + 10% olive cake, two phases; T4—base diet + 10% exhausted olive cake; T5—base diet + 10% exhausted olive cake + 1% olive oil. ** *p* < 0.01; * *p* < 0.05. Mean values with different lowercase letters within the same parameter are significantly different (for the treatments).

**Table 3 animals-14-01697-t003:** Physical–chemical composition in Bísaro pork dry-cured shoulder. Effect of muscles (infraspinatus, supraspinatus, and subscapularis), and effect of time of curing process (raw meat, 6 months of curing, and 12 months of curing).

Muscle	Time Curing	Chemical Composition (g/100g)						
		aw	Moisture	NaCl	Ash	Protein	Total Fat	Collagen
SB	RM	0.960 a	72.19 a		1.73 e	19.12 f	6.34 f	1.62 bcd
	6MC	0.896 b	42.27 d	3.75 c	4.82 d	34.29 b	18.71 ab	0.67 d
	12MC	0.805 d	33.24 e	3.81 c	5.38 cd	39.99 a	23.02 a	3.62 a
IF	RM	0.954 a	71.10 a		1.49 e	18.74 f	7.99 def	1.64 bc
	6MC	0.908 b	52.75 c	4.32 bc	5.83 c	23.64 e	11.28 cde	1.64 bcd
	12MC	0.841 c	43.99 d	6.09 a	7.08 ab	30.00 c	13.97 bc	4.36 a
SP	RM	0.963 a	71.88 a		1.65 e	19.20 f	7.29 ef	0.94 cd
	6MC	0.915 b	59.28 b	4.96 b	6.16 bc	26.58 d	6.32 f	0.84 cd
	12MC	0.845 c	50.32 c	6.13 a	7.15 a	31.81 bc	12.06 cd	2.48 b
SEM		0.005	1.072	0.221	0.210	0.587	1.115	0.246
significance		***	***	***	***	***	***	**

SB—subcapularis; IF—infraspinatus; SP—supraspinatus; *** *p* < 0.001; ** *p* < 0.01. Mean values with different lowercase letters within the same parameter are significantly different.

**Table 4 animals-14-01697-t004:** Effect of inclusion of olive cake into the animal diet in fatty acid profile in Bísaro pork dry-cured shoulder.

	Treatment						
Fatty Acids (g/100g)	T1	T2	T3	T4	T5	SE	*p*
C16:0	24.30	24.46	24.13	24.58	24.23	0.14	ns
C16:1n-7	2.74	2.83	2.73	2.85	2.78	0.051	ns
C18:0	11.42	11.40	11.47	11.29	11.27	0.146	ns
C18:1n-9	47.99 ab	47.76 b	48.78 a	47.85 ab	48.31 ab	0.294	*
C18:2n-6	8.32	8.56	8.22	8.58	8.58	0.162	ns
SFA	37.70	37.79	37.53	37.86	37.50	0.25	ns
MUFA	52.16	52.42	52.81	52.07	52.43	0.23	ns
PUFA	10.14	9.79	9.66	10.07	10.07	0.20	ns
n-6/n-3	24.42	24.26	24.53	24.17	24.16	0.68	ns
IA	0.46	0.46	0.46	0.47	0.46	0.004	ns
IT	1.14	1.15	1.14	1.15	1.13	0.012	ns
h/H	2.27	2.24	2.31	2.24	2.29	0.022	ns

T1—base diet and commercial feed; T2—base diet + 10% crude olive cake; T3—base diet + 10% olive cake, two phases; T4—base diet + 10% exhausted olive cake; T5—base diet + 10% exhausted olive cake + 1% olive oil. * *p* < 0.05; ns—not significant. SFAs, saturated fatty acids; MUFAs, monounsaturated fatty acids; PUFAs, polyunsaturated fatty acids; PUFA, n-6/n-3 (∑omega-6)/(∑omega-3); IA, index of atherogenicity; IT, index of thrombogenicity; h/H = (C18: 1n − 9 + C18: 2n − 6 + C20: 4n − 6 + C18: 3n − 3 + C20; 5 − n3 + C22: 5n − 3 + C22: 6n − 3)/C14: 0 + C16: 0; only the most representative fatty acids are considered in this table. Mean values with different lowercase letters within the same parameter are significantly different.

**Table 5 animals-14-01697-t005:** Effect of muscle and time curing in fatty acid profile in Bísaro pork dry-cured shoulder.

Fatty Acids	Muscle × Time of Curing										
	SB			IF			SP			SEM	SIG.
	RM	6MC	12MC	RM	6MC	12MC	RM	6MC	12MC		
C10:0	0.014 c	0.024 bc	0.031 abc	0.046 a	0.031 abc	0.026 bc	0.040 ab	0.026 bc	0.025 bc	0.005	***
C12:0	0.026 d	0.029 bcd	0.047 abc	0.054 a	0.032 bcd	0.041 abcd	0.045 ab	0.024 cd	0.035 abcd	0.005	***
C14:0	1.06 bc	1.19 a	1.11 abc	1.06 bc	1.14 ab	1.05 bc	1.08 bc	1.13 ab	1.04 cd	0.020	*
C14:1	0.008 cd	0.004 d	0.020 abc	0.031 a	0.004 d	0.022 ab	0.016 bcd	0.005 d	0.02 abc	0.003	***
C15:0	0.32 d	0.12 bcd	0.05 d	0.07 a	0.14 cd	0.11	0.22 bc	0.25 ab	0.12 bcd	0.029	***
C16:0	23.73 d	24.71 bc	26.01 a	24.72 b	23.86 cd	24.73 bc	23.65 d	23.86 cd	24.69 bc	0.206	***
C16:1n-7	2.72 b	2.95 ab	2.66 b	2.68 b	2.92 ab	2.66 b	2.80 b	3.11 a	2.72 b	0.074	ns
C17:0	0.23 a	0.22 a	0.14 bc	0.22 a	0.21 a	0.11 c	0.21 a	0.19 ab	0.12 c	0.013	ns
C17:1n-7	0.23 a	0.22 a	0.22 a	0.25 a	0.26 a	0.23 a	0.25 a	0.24 a	0.22 a	0.013	ns
C18:0	11.33 bc	11.20 bc	12.77 a	11.88 b	10.58 cd	11.78 b	10.89 cd	10.16 d	11.77 b	0.215	***
9t-C18:1	0.14 a	0.17 a	1.38 a	0.17 a	0.18 a	0.20 a	0.16 a	0.16 a	0.19 a	0.331	ns
C18:1n-9	47.75 bcd	49.02 ab	46.57 d	47.28 cd	49.88 a	48.54 abc	48.50 abc	48.74 abc	47.85 bcd	0.432	**
9t,12t-C18:2	0.009 b	0.011 ab	0.011 ab	0.020 a	0.014 ab	0.011 ab	0.009 b	0.017 ab	0.010 ab	0.003	ns
C18:2n-6	9.33 a	7.68 de	6.89 e	8.54 bcd	7.94 cde	7.89 cde	8.97 ab	8.83 abc	8.47 abcd	0.238	***
C20:0	0.13 ab	0.14 ab	0.16 a	0.16 a	0.15 a	0.16 a	0.15 a	0.12 b	0.14 ab	0.008	**
C18:3n-6	0.010 a	0.004 a	0.028 a	0.027 a	0.010 a	0.015 a	0.023 a	0.022 a	0.017 a	0.008	ns
C20:1n-9	0.80 ab	0.80 ab	0.68 c	0.79 ab	0.86 a	0.77 abc	0.80 ab	0.73 bc	0.73 bc	0.024	*
C18:3n-3	0.34 ab	0.30 bcd	0.21 e	0.32 ab	0.37 a	0.27 d	0.31 bc	0.35 ab	0.28 cd	0.012	***
C21:0	0.031 abc	0.015 bc	0.008 bc	0.044 a	0.033 abc	0.012 bc	0.034 ab	0.015 bc	0.005 c	0.007	ns
C20:2n-6	0.34 a	0.29 b	0.31 ab	0.32 ab	0.31 ab	0.33 ab	0.34 ab	0.30 ab	0.33 ab	0.013	ns
C22:0	0.02 c	0.02 c	0.03 bc	0.06 a	0.03 bc	0.05 ab	0.05 ab	0.04 abc	0.05 ab	0.005	ns
C20:3n-6	0.13 ab	0.08 cd	0.06 d	0.12 ab	0.11 abc	0.10 bcd	0.14 a	0.14 a	0.10 bc	0.008	***
C20:3n-3	0.03 b	0.01 bc	0.01 bc	0.08 a	0.02 bc	0.001 c	0.08 a	0.02 bc	0.004 bc	0.007	***
C20:4n-6	1.02 a	0.63 bc	0.001 d	0.42 c	0.69 b	0.003 d	0.98 a	1.22 a	0.001 d	0.062	***
C24:1n-9	0.18 a	0.09 de	0.07 e	0.16 ab	0.12 bcd	0.12 cde	0.18 a	0.16 abc	0.11 bcde	0.010	***
C22:6n-3	0.06 ab	0.02 d	0.03 cd	0.06 ab	0.05 bc	0.05 abc	0.08 a	0.06 ab	0.05 bc	0.007	ns
SFA	36.90 cd	37.69 bc	40.78 a	38.71 b	36.23 cd	38.74 b	36.37 cd	35.88 d	38.84 b	0.361	***
MUFA	51.84 cd	53.26 ab	51.64 cd	51.36 d	54.24 a	52.58 bcd	52.69 bc	53.14 ab	51.88 bcd	0.335	***
PUFA	11.26 a	9.05 cd	7.58 e	9.93 bc	9.52 cd	8.68 de	10.94 a	10.98 ab	9.28 cd	0.291	***
n-6/n-3	26.40 a	26.62 a	28.06 a	20.24 d	20.83 cd	26.13 ab	22.32 bcd	25.24 abc	26.89 ab	1.005	ns
IA	0.44 e	0.47 bc	0.52 a	0.47 b	0.45 cde	0.48 bc	0.44 e	0.44 de	0.47 bcd	0.007	***
IT	1.11 cd	1.16 bc	1.32 a	1.18 b	1.08 d	1.19 b	1.08 d	1.06 d	1.19 b	0.018	***
h/H	2.36 a	2.23 bc	1.98 d	2.20 c	2.36 ab	2.21 c	2.38 a	2.37 ab	2.20 c	0.033	***

ns—not significant, * *p* < 0.05, ** *p* < 0.01, *** *p* < 0.001; SFAs, saturated fatty acids; MUFAs, monounsaturated fatty acids; PUFAs, polyunsaturated fatty acids; PUFA, n-6/n-3 (∑omega-6)/(∑omega-3); IA, index of atherogenicity; IT, index of thrombogenicity; h/H = (C18: 1n − 9 + C18: 2n − 6 + C20: 4n − 6 + C18: 3n − 3 + C20; 5 − n3 + C22: 5n − 3 + C22: 6n − 3)/C14: 0 + C16: 0; only fatty acids that represented more than 0.1% are presented in the table, although all detected fatty acids were used for calculating the totals and the indices. SB—subscapularis; IF—infraspinatus; SP—supraspinatus; RM—raw meat; 6MC—6 months of curing; 12MC—12 months of curing. Mean values with different lowercase letters within the same parameter are significantly different.

**Table 6 animals-14-01697-t006:** *F*—values in the discriminant analysis.

Variable	F Ratio	Prob > F
aw	20.508	0.0000000
Moisture	14.631	0.0000000
Ash	38.971	0.0000000
Total Fat	5.979	0.0000000
Protein	20.587	0.0000000
Collagen	12.769	0.0000000
SFA	9.448	0.0000000
n-6/n-3	6.384	0.0000000

## Data Availability

All data were presented in the manuscript. Data can be requested from the corresponding author via email.

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
