# Peer review of "Effect of Olive Cake in Bísaro Pig Feed on Physicochemical Composition and Fatty Acid Profile of Three Different Muscles of Dry-Cured Shoulder"

_animals, 2024, doi:10.3390/ani14111697_

Round 1
Reviewer 1 Report
Comments and Suggestions for Authors
Dears, below are some suggestions to improve the quality of the manuscript.
Delete what is marked in red.
107-111 - The experimental feed trial was carried out during the finishing phase, for a total of 3 months, at Trás-os-Montes e Alto Douro University, Vila Real, Portugal. Analysis of diets was performed at the Meat Technology Center of Galicia, Ourense, Spain. Animals were harvested when they reached 12 months of age, at around 135 kg live weight and 110 kg carcass weight, at the Municipal Slaughterhouse of Bragança.
161 – Portuguese Institute of Quality (NP 1614/2002).
Please follow the same for all other referred methodologies.
161-164 – to?? Are you trying to say Two?
Maybe: In duplicate, three (3) g of sample was added to 5 mL of ethanol…
169 – were
174 – what method number?
179 –180 at the Carcass and Meat Quality…... following the method proposed by Folch [24]. The total lipids..
185 – 2 mL of distilled water was added, and samples were vortexed again.
195, 394, and 461 – atherogenic index
195, 394, and 462 – thrombogenic index
223 – As reported in table 2,…
280 - ????
281 – As noted/observed in Table 3, the….
374 – most abundant?
375 – fatty acid
376 – The same as other products
389 – Table 4 – values also reported on g/100g?
How h/H was obtained? Is this reported on material and method?
413 – fatty acid
418 – rese?
419 – Regarding IF muscle, ….
502 – As mentioned before, this muscle
512 – What’s more??????
Rephrase
573 – Too extensive and redundant
579 – once it uses a co-product from the olive industry
583 – significantly different chemical composition values.
593 – does not affect

Comments on the Quality of English LanguageEnglish requires some review.
Author Response
Dear review,
All modifications were made following the reviewer's suggestions and comments, and responses to their comments are also attached. Thanks to their recommendations, significant modifications were made throughout the manuscript.
Thank you for your attention.
Answers to Reviewer 1
Dears, below are some suggestions to improve the quality of the manuscript.
Delete what is marked in red.
Response: We agree. They´ve been removed.
107-111 - The experimental feed trial was carried out during the finishing phase, for a total of 3 months, at Trás-os-Montes e Alto Douro University, Vila Real, Portugal. Analysis of diets was performed at the Meat Technology Center of Galicia, Ourense, Spain. Animals were harvested when they reached 12 months of age, at around 135 kg live weight and 110 kg carcass weight, at the Municipal Slaughterhouse of Bragança.
Response: We agree. The changes have been made.
161 – Portuguese Institute of Quality (NP 1614/2002).
Please follow the same for all other referred methodologies.
Response: We agree. The changes have been made.
161-164 – to?? Are you trying to say Two?
Maybe: In duplicate, three (3) g of sample was added to 5 mL of ethanol…
Response: We agree. The changes have been made.
169 – were
Response: We agree. The changes have been made.
174 – what method number?
Response: We agree. The changes have been made.
179 –180 at the Carcass and Meat Quality…... following the method proposed by Folch [24]. The total lipids..
Response: We agree. The changes have been made.
185 – 2 mL of distilled water was added, and samples were vortexed again.
Response: We agree. The changes have been made.
195, 394, and 461 – atherogenic index
Response: We agree. The changes have been made.
195, 394, and 462 – thrombogenic index
Response: We agree. The changes have been made.
223 – As reported in table 2,…
Response: We agree. The changes have been made.
280 - ????
Response: We agree. The changes have been made.
281 – As noted/observed in Table 3, the….
Response: We agree. The changes have been made.
374 – most abundant?
Response: We agree. The changes have been made.
375 – fatty acid
Response: We agree. The changes have been made.
376 – The same as other products
Response: We agree. The changes have been made.
389 – Table 4 – values also reported on g/100g?
Response: g/100g. Added to the table
How h/H was obtained? Is this reported on material and method?
?/ ? = (C18: 1n − 9 + C18: 2n − 6 + C20: 4n − 6 + C18: 3n − 3 + C20; 5 − n3 + C22: 5n − 3 + C22: 6n – 3)/ C14: 0 + C16: 0
Response: the calculation formula has been included in table 4 and 5
413 – fatty acid
Response: We agree. The changes have been made.
418 – rese?
Response: research. The changes have been made.
419 – Regarding IF muscle, ….
Response: We agree. The changes have been made.
502 – As mentioned before, this muscle
Response: We agree. The changes have been made.
512 – What’s more??????
Rephrase
Response: We agree. The changes have been made.
573 – Too extensive and redundant
Response: We agree. The changes have been made.
579 – once it uses a co-product from the olive industry
Response: We agree. The changes have been made.
583 – significantly different chemical composition values.
Response: We agree. The changes have been made.
593 – does not affect
Response: We agree. The changes have been made.

Reviewer 2 Report
Comments and Suggestions for Authors
Title: Effect of olive cake in Bísaro pig feed on the physicochemical composition and fatty acid profile of three different muscles of dry-cured shoulder
The manuscript “Effect of olive cake in Bísaro pig feed on the physicochemical composition and fatty acid profile of three different muscles of dry-cured shoulder” assessed the following effects: (1) inclusion of olive in the animal´s diet on dry-cured shoulder; (2) the effect of curing on three different muscles (infraspinatus, supraspinatus and subscapular); (3) the effect of different curing times (fresh shoulder, 6 months curing and 12 months curing). The study concluded that the introduction of olive cake into the animal diet does not negatively influence the final product obtained and that this by-product with an environmentally high toxic level can be used, contributing to a sustainable circular economy. It is a well-written article with some interesting findings; however, there are some corrections before its acceptance for publication:
Line 24-31: In the abstract portion authors should mention the P-value along with the significant results and I would suggest omitting the insignificant results from this portion.
Line 26: What was the treatment T4? Authors should mention any abbreviation after describing it.
Line 30-31: What do the SFA, MUFA and PUFA stand for?
Line 34: There must be a concluding sentence at the end of the abstract to make it meaningful for the readers.
Line 46: Clarify the term no-existent value of olive cake. Perhaps use low value instead.
Line 55-57: Specify how incorporating olive cake into pig feed contributes to environmental sustainability.
Line 58-59: Clarify what is meant by a possible alternative to typical/commercial basic diets.
Line 38-90: I would suggest the authors split the introduction part/sentences into several parts for better understanding of the readers.
Line 94: Specify what is meant by an extensive system in pig farming.
Lines 99-104: Provide more details about the different treatments, such as the specific differences between them and the rationale behind each treatment.
Line 114: At what postmortem time the samples were collected and at what temperature they were stored until analysis?
Line 125: Specify why the shoulders were cured for 12 months and why this duration was chosen.
Lines 158-177: Provide more details about the procedures used for the chemical composition and physicochemical analysis, including the specific methods and equipment used.
Line 221: It should be “Results and Discussion”
Line 223: Consider starting this section with a brief overview sentence summarizing the main findings regarding the influence of olive cake treatments on the physicochemical composition.
Line 230-231: Consider rephrasing to improve clarity: “In the same study [28], values similar to those obtained by us were observed for collagen in dry-cured loin (1.59-2.84%) and the dry-cured cachaco (1.54-2.09%)”
Lines 261-263: This sentence could be clearer: "The average aw values obtained in all treatments were practically identical (0.887), indicating that the inclusion of olive cake did not affect the drop in aw value during the curing process."
Line 370-474: Use more concise language to describe the trends in fatty acid composition.
My overall comments on the results and discussion part are:
· Clarify the comparison between your results and those of other studies for better understanding.
· Consider simplifying complex sentences for easier comprehension.
Line 569: The conclusion part is very lengthy and the authors included the general discussion in it, they may be moved such information to the introduction part of the manuscript. Therefore, I would suggest concluding the study based on the results and drawing some futuristic approaches for the scientists, meat processors or farmers.
Comments on the Quality of English LanguageEnglish grammar and sentence structure should be revised and corrected throughout the manuscript.
Author Response
Dear review,
All modifications were made following the reviewer's suggestions and comments, and responses to their comments are also attached. Thanks to their recommendations, significant modifications were made throughout the manuscript.
Thank you for your attention.
Answers to Reviewer 2
The manuscript “Effect of olive cake in Bísaro pig feed on the physicochemical composition and fatty acid profile of three different muscles of dry-cured shoulder” assessed the following effects: (1) inclusion of olive in the animal´s diet on dry-cured shoulder; (2) the effect of curing on three different muscles (infraspinatus, supraspinatus and subscapular); (3) the effect of different curing times (fresh shoulder, 6 months curing and 12 months curing). The study concluded that the introduction of olive cake into the animal diet does not negatively influence the final product obtained and that this by-product with an environmentally high toxic level can be used, contributing to a sustainable circular economy. It is a well-written article with some interesting findings; however, there are some corrections before its acceptance for publication:
Line 24-31: In the abstract portion authors should mention the P-value along with the significant results and I would suggest omitting the insignificant results from this portion.
Response: Since the p-values (*) ≤ 0.05; (**) ≤ 0.01 and (***) ≤ 0.001 were entered, we kept the same logic in the abstract. We remove what we consider to be the most insignificant results.
line 26: What was the treatment T4? Authors should mention any abbreviation after describing it.
Response: We agree. The information has been added to the text
Line 30-31: What do the SFA, MUFA and PUFA stand for?
Response: The information has been added to the text
Line 34: There must be a concluding sentence at the end of the abstract to make it meaningful for the readers.
Response: The information has been added to the text
Line 46: Clarify the term no-existent value of olive cake. Perhaps use low value instead.
Response: We agree. The information has been added to the text.
Line 55-57: Specify how incorporating olive cake into pig feed contributes to environmental sustainability.
Response: The information has been added to the text.
Line 58-59: Clarify what is meant by a possible alternative to typical/commercial basic diets.
Response: The information has been added to the text.
Line 38-90: I would suggest the authors split the introduction part/sentences into several parts for better understanding of the readers.
Response: We agree. The information has been added to the text.
Line 94: Specify what is meant by an extensive system in pig farming.
Response: The information has been added to the text.
Lines 99-104: Provide more details about the different treatments, such as the specific differences between them and the rationale behind each treatment.
Response: The information has been added to the text (line 114-118; 122-124)
Line 114: At what postmortem time the samples were collected and at what temperature they were stored until analysis?
Response: The information has been added to the text (line 134-139)
Line 125: Specify why the shoulders were cured for 12 months and why this duration was chosen.
Response: The curing process was carried out according to the manufacturing flowchart for Bísaro pork shoulder from the company Bísaro-Salsicharia Tradicional, Lda ®. We appreciate the suggestion, but we don´t think that adding this information will do anything for the reader. We would also add that this company has been standardized for more than 7 years with very demanding quality references (such as the IFS-Food reference). For this reason, we have chosen no to change steps, times and/or temperatures in the manufacturing flowchart for this product.
Lines 158-177: Provide more details about the procedures used for the chemical composition and physicochemical analysis, including the specific methods and equipment used.
Response: We appreciate the suggestion, but we believe that the procedures include all the necessary information, including the type of equipment used and methodologies. In any case, we´re available to change any specific points you indicate.
Line 221: It should be “Results and Discussion”
Response: We agree. The information has been added to the text.
Line 223: Consider starting this section with a brief overview sentence summarizing the main findings regarding the influence of olive cake treatments on the physicochemical composition.
Response: We agree. The information has been added to the text.
Line 230-231: Consider rephrasing to improve clarity: “In the same study [28], values similar to those obtained by us were observed for collagen in dry-cured loin (1.59-2.84%) and the dry-cured cachaco (1.54-2.09%)”
Response: We agree. The information has been added to the text.
Lines 261-263: This sentence could be clearer: "The average aw values obtained in all treatments were practically identical (0.887), indicating that the inclusion of olive cake did not affect the drop in aw value during the curing process."
Response: The information has been added to the text.
Line 370-474: Use more concise language to describe the trends in fatty acid composition.
Response: We agree. The information has been added to the text.
My overall comments on the results and discussion part are:
- Clarify the comparison between your results and those of other studies for better understanding.
- Consider simplifying complex sentences for easier comprehension.
Response: We agree. The information has been added to the text.
Line 569: The conclusion part is very lengthy and the authors included the general discussion in it, they may be moved such information to the introduction part of the manuscript. Therefore, I would suggest concluding the study based on the results and drawing some futuristic approaches for the scientists, meat processors or farmers.
Response: We agree. The information has been added to the text.

Round 2
Reviewer 1 Report
Comments and Suggestions for Authors
Significant changes were addressed to improve the quality of the manuscript.
Comments on the Quality of English LanguageMinor revisions.
Reviewer 2 Report
Comments and Suggestions for Authors
The manuscript is sufficiently improved and may be accepted in its present form for possible publication in Animals.